# Biosynthesis of ilamycins featuring unusual building blocks and engineered production of enhanced anti-tuberculosis agents

Junying Ma[1], Hongbo Huang[1], Yunchang Xie[1], Zhiyong Liu[2], Jin Zhao[3], Chunyan Zhang[1,4], Yanxi Jia[1,4], Yun Zhang[1], Hua Zhang[3], Tianyu Zhang [2] & Jianhua Ju[1,4]

Tuberculosis remains one of the world's deadliest communicable diseases, novel anti-tuberculosis agents are urgently needed due to severe drug resistance and the co-epidemic of tuberculosis/human immunodeficiency virus. Here, we show the isolation of six anti-mycobacterial ilamycin congeners (**1–6**) bearing rare L-3-nitro-tyrosine and L-2-amino-4-hexenoic acid structural units from the deep sea-derived *Streptomyces atratus* SCSIO ZH16. The biosynthesis of the rare L-3-nitrotyrosine and L-2-amino-4-hexenoic acid units as well as three pre-tailoring and two post-tailoring steps are probed in the ilamycin biosynthetic machinery through a series of gene inactivation, precursor chemical complementation, isotope-labeled precursor feeding experiments, as well as structural elucidation of three intermediates (**6–8**) from the respective mutants. Most impressively, ilamycins $E_1/E_2$, which are produced in high titers by a genetically engineered mutant strain, show very potent anti-tuberculosis activity with an minimum inhibitory concentration value ≈9.8 nM to *Mycobacterium tuberculosis* H37Rv constituting extremely potent and exciting anti-tuberculosis drug leads.

[1] CAS Key Laboratory of Tropical Marine Bio-Resources and Ecology, Guangdong Key Laboratory of Marine Materia Medica, RNAM Center for Marine Microbiology, South China Sea Institute of Oceanology, Chinese Academy of Sciences, Guangzhou 510301, China. [2] Tuberculosis Research Laboratory, State Key Laboratory of Respiratory Disease, Guangzhou Institutes of Biomedicine and Health, Chinese Academy of Sciences, Guangzhou 510530, China. [3] Guangdong Provincial Key Laboratory of Medical Molecular Diagnostics, Institute of Laboratory Medicine, Guangdong Medical University, Dongguan 523808, China. [4] University of Chinese Academy of Sciences, Beijing 100049, China. Correspondence and requests for materials should be addressed to J.M. (email: majunying@scsio.ac.cn) or to J.J. (email: jju@scsio.ac.cn)

Tuberculosis (TB) ranks as the top infectious killer in the world, with the number of TB deaths exceeding those from human immunodeficiency virus (HIV)[1]. It is estimated that two billion people—one-third of the world's population—have latent TB, and about 9.6 million people fall ill; 1.5 million people die from TB annually[2, 3]. Moreover, the prevention and control of TB have become more difficult because of the co-epidemic of TB/HIV as well as the emergence and rapid dissemination of multidrug-resistant, extensively drug-resistant, and totally drug-resistant strains[4, 5]. Hence, novel anti-TB agents with increased potency and efficacy are urgently needed.

During the course of our efforts to discover and engineer anti-infective and anti-tumor agents from marine-derived actinomycetes[6–8], six compounds with the ultraviolet (UV) spectrum of 220, 285, and 352 nm were obtained from a deep South China Sea-derived strain *Streptomyces atratus* SCSIO ZH16. The high-resolution mass spectrometry (HRMS) profiles and nuclear magnetic resonance (NMR) spectroscopic data characterized them to be ilamycins bearing two rare units of L-3-nitrotyrosine and L-2-amino-4-hexenoic acid (L-AHA), which was originally isolated as single, or as mixtures of, components from several *Streptomyces* in the early 1960s–1970s[9–11], and then re-isolated as rufomycins in another *S. macrosporeus* DSM-12818 in 2000[12–14]. The structures, including most of the stereochemistry, of five ilamycins B₁ (**1**), B₂ (**2**), C₁ (**3**), C₂ (**4**), and D (**5**) were characterized by HRMS, 1D and 2D NMR spectroscopic data analyses, and chemical transformations[12–14]. Notably, the biological activities of ilamycins B₁, B₂, C₂, and D were not reported; only ilamycin C₁ was noted to display inhibitory activity against *Mycobacterium tuberculosis* H37Rv (minimum inhibitory concentration (MIC) < 1.3 µg mL⁻¹)[13]. However, in all previous publications, the absolute configurations of the epoxy groups in ilamycins B₂, C₁, C₂, and D, as well as the γ-C (C₃₂) of the 2-amino-4-methylpentanedioic acid unit in ilamycin D were still unsolved.

Recently, the repurposing of old drugs and the re-evaluation of natural product leads have become new approaches to identifying anti-TB drug candidates[15–17]. Given the intriguing anti-mycobacterial activities that had been reported for the ilamycins, the still unresolved questions about absolute stereochemistries, and the lack of systematic studies and precise knowledge about the biological activities of each of the purified compounds, a thorough and in-depth investigation of the chemistry and biology of the full set of ilamycins was clearly warranted. Furthermore, elucidation of the biosynthetic pathway to these interesting agents was deemed necessary; the ultimate goal envisioned entails ilamycin engineering to generated analogs with improved anti-mycobacterial potencies.

In this paper, we isolate six ilamycins (B₁, B₂, C₁, C₂, D, and E₁; **1**–**6**) (Fig. 1) from a marine-derived *S. atratus* SCSIO ZH16 strain (Fig. 2, trace i) and determine the absolute stereochemistries of

the epoxide groups (C₁₃) and the γ-C (C₃₂) of the 2-amino-4-methylpentanedioic acid residue by X-ray diffraction analysis of ilamycins B₂, C₂, and D. We also identify and analyze the ilamycin biosynthetic gene cluster in *S. atratus* SCSIO ZH16. In addition, the biosynthetic routes to two rare natural product building blocks, L-3-nitrotyrosine and L-AHA, are determined via gene inactivation, isotope-labeled precursor feeding, and chemical complementation experiments. Moreover, we elucidate the pre-tailoring and post-tailoring steps in ilamycin biosynthesis and obtain three ilamycins analogs (E₁, E₂, and F; **6**–**8**, Fig. 1). Finally, we evaluate eight ilamycins (**1**–**8**) for anti-mycobacterial activities against *M. smegmatis* MC² 155 and *M. tuberculosis* H37Rv and cytotoxic activities against a panel of human tumor and normal cell lines. These last studies revealed two engineered ilamycins as anti-TB drug leads with potencies in the nanomolar range.

## Results

**Discovery and structural elucidation of ilamycins.** The strain SCSIO ZH16 was isolated from a deep South China Sea sediment sample and identified as *Streptomyces atratus* by morphology and 16S DNA sequence analyses. Subsequent large-scale fermentation (16L), extraction, and careful isolation by silica gel column chromatography (CC) followed by preparative high performance liquid chromatography (HPLC) afforded analytically pure compounds **1**–**6**. The structures of compounds **1**–**5** were identified and designated as ilamycins B₁ (**1**), B₂ (**2**), C₁ (**3**), C₂ (**4**), and D (**5**), respectively, by HRMS (Supplementary Figs. 35–37) and 1D and 2D (correlation spectroscopy (COSY), heteronuclear single-quantum correlation spectroscopy (HSQC), and heteronuclear multiple-bond correlation spectroscopy (HMBC)) NMR data (Supplementary Figs. 42, 43, 45–54) analyses and by comparison with previously reported data for the ilamycins (or rufomycins)[9–14].

In this study, after careful incubation in suitable solvents, qualified single crystals of ilamycin B₂ in MeOH, ilamycin C₂ in MeOH-CHCl₃ (9:1), and ilamycin D in MeOH-EtOH (1:1) were obtained for X-ray diffraction. We subsequently collected X-ray diffraction data for ilamycin B₂, C₂, and D (Supplementary Table 1; Supplementary Data 1–3, 5). Analysis of the X-ray diffraction data confirmed the previous structures that had been deduced by spectroscopic analysis and chemical derivatization (Supplementary Figs. 1–3)[9–14]. In addition, the crystal of ilamycin C₂ (**4**) contains two CHCl₃ molecules inserted into the ilamycin C₂ molecule in a unit cell, enabling the stereochemistries within the six-membered ring at C₃₂ and C₃₃ both be established as S (absolute structure parameter = −0.012(3)). The single-crystal X-ray data of ilamycin C₂ also revealed that the conformation of the hemiaminal-containing six-membered ring adopts a twist boat conformation vertical to the macrolactam ring. Analysis of the X-ray diffraction data for ilamycin B₂ (**2**) and ilamycin D (**5**), together with the L-configurations established for all amino acid

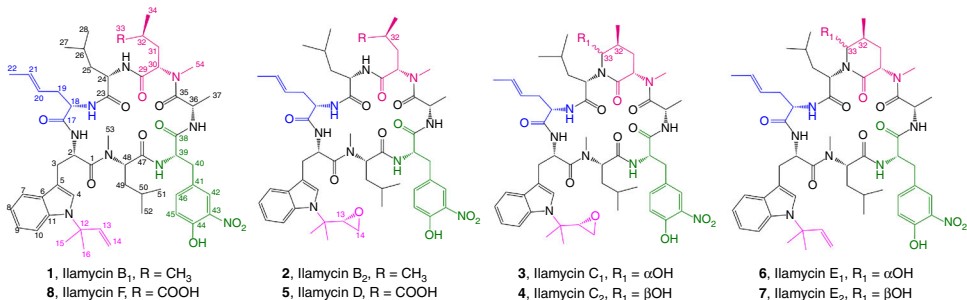

**Fig. 1** Structures of ilamycins. Compounds **1**–**6** were isolated from *S. atratus* SCSIO ZH16 wild-type strain, compounds **6**–**8** were isolated from engineered mutant strains

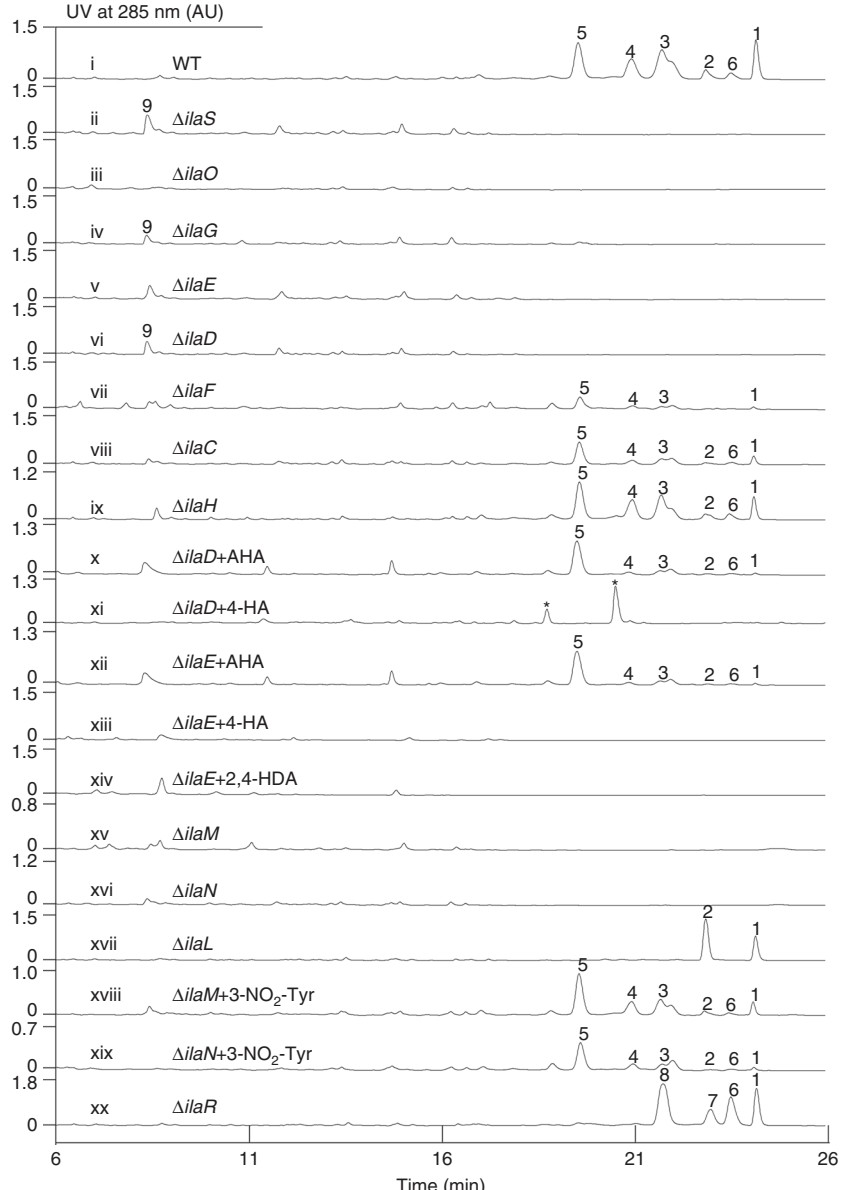

**Fig. 2** HPLC analyses of fermentation broths. (i) wild-type *S. atratus* SCSIO ZH16; (ii) Δ*ilaS* mutant; (iii) Δ*ilaO* mutant; (iv) Δ*ilaG* mutant; (v) Δ*ilaE* mutant; (vi) Δ*ilaD* mutant n; (vii) Δ*ilaF* mutant; (viii) Δ*ilaC* mutant; (ix) Δ*ilaH* mutant; (x) Δ*ilaD* mutant fed with AHA; (xi) Δ*ilaD* mutant fed with 4-HA; (xii) Δ*ilaE* mutant fed with AHA; (xiii) Δ*ilaE* mutant fed with 4-HA; (xiv) Δ*ilaE* mutant fed with 2,4-HDA; (xv) Δ*ilaM* mutant; (xvi) Δ*ilaN* mutant; (xvii) Δ*ilaL* mutant; (xviii) Δ*ilaM* mutant fed with 3-NO$_2$-tyr; (xix) Δ*ilaN* mutant fed with 3-NO$_2$-tyr; (xx) Δ*ilaR* mutant; the peaks labeled with *asterisks* are not ilamycin analogs judged by HPLC–DAD–UV analysis

residues in these heptapeptides[10], determined the *S* configurations for C$_{13}$ and C$_{32}$ in **2** and **5**.

Minor product **6** was purified as a yellow powder. The molecular formula of **6** was established by high resolution electrospray ionization massspectroscopy (HRESIMS) to be C$_{54}$H$_{75}$N$_9$O$_{11}$, 16 mass units less than that of ilamycin C$_1$. Comparison of the $^1$H and $^{13}$C NMR spectroscopic data (Supplementary Table 2; Supplementary Figs. 55, 56) of **6** with that of ilamycin C$_1$ revealed the absence of the signals corresponding to the epoxy group of ilamycin C$_1$ (δ$_C$ 59.7, δ$_H$ 3.26, CH$_2$-13; δ$_C$ 46.0, δ$_H$ 2.88, 2.83, CH$_2$-14), and the appearance of a new set of signals consistent with a terminal olefin at δ$_C$ 145.5, δ$_H$ 6.15 (CH-13) and δ$_C$ 114.1, δ$_H$ 5.22, 5.19 (CH2-14), suggesting that the epoxy group of ilamycin C$_1$ was replaced by a terminal olefin in **6**. Further 2D NMR (Supplementary Figs. 57–59) analysis confirmed the presence of an isopentenyl group in **6**,

which is also present in ilamycin B$_1$ (**1**). Comparison of the $^1$H and $^{13}$C NMR data of **6**, especially the chemical shifts at C$_{32}$ and C$_{33}$, also suggest that **6** has the same stereochemistries as ilamycin C$_1$ (**3**);[12–14] compound **6** was subsequently designated ilamycin E$_1$.

**Identification of the ilamycins gene cluster.** The presence of non-proteingenic amino acids in the ilamycin cyclopeptide backbone supported our hypothesis that ilamycin biosynthesis is governed by nonribosomal peptide synthetases (NRPSs). Accordingly, we sought to identify the *ila* gene cluster by whole-genome sequencing of *S. atratus* SCSIO ZH16 using a combination of second-generation 454 and third-generation PacBio sequencing technology. The size of the entire linear genome of *S. atratus* SCSIO ZH16 is 9.64 Mbp. Upon data annotation and

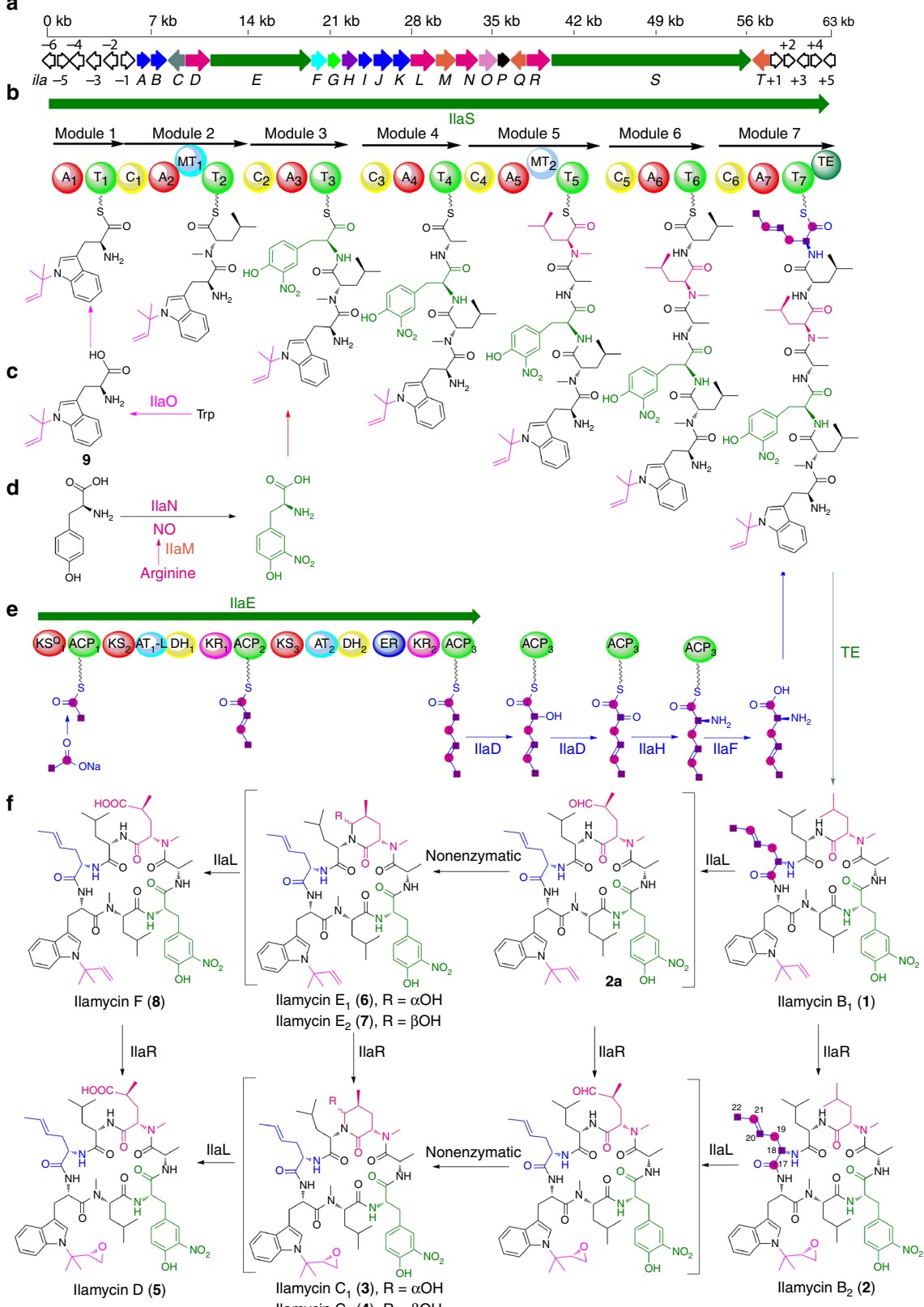

**Fig. 3** Biosynthetic gene cluster and proposed biosynthetic pathway of ilamycins. **a** Organization of the ilamycin gene cluster. **b** Biosynthetic pathway of ilamycins. **c** The prenylation of Trp. **d** The nitration of Tyr. **e** The biosynthesis of L-AHA unit. **f** The post-tailoring biosynthetic steps en route to ilamycins. *A* adenylation, *C* condensation, *T* thiolation, *MT* methylation, *TE* thioesterase, *KS* keto synthases, *ACP* acyl carrier protein, *AT* acyl transferase, *AT-L* acyl transferase-like protein, *DH* dehydratase, *KR* keto reductase, *ER* enoyl reductase

analysis with antiSMASH[18], a *ca* 57.1 kb DNA segment consisting of 20 open reading frames was identified to likely be involved in the biosynthesis of ilamycins (Supplementary Table 4). The nucleotide sequences have been deposited in GenBank with accession number KY173348, and the gene cluster is shown in Fig. 3a. A cosmid library of *S. atratus* SCSIO ZH16 was constructed using the SuperCos1 vector system and 12 positive clones were screened for gene inactivations.

Consistent with the cyclic peptide scaffold of the ilamycins, we found a giant 8022 aa protein, IlaS, comprising seven modules that incorporate building blocks to form the full-length heptapeptide (Fig. 3b). Polyketide synthase (PKS)/NRPS analyses employing online software revealed that the predicted substrate amino acids of the A1–A7 binding pocket domain of IlaS do not match those found in the ilamycins. To confirm that the analyzed gene cluster is responsible for ilamycin biosynthesis, we constructed a Δ*ilaS* mutant by using λ-Red recombination with an apramycin gene cassette[19–21]. The mutant was identified and confirmed by its kanamycin-sensitive and apramycin-resistant phenotype and further validated by polymerase chain reaction (PCR). The Δ*ilaS* mutant completely lost the ability to produce ilamycins but the HPLC profile showed accumulation of compound **9** (Fig. 2, trace ii). Compound **9** was isolated upon large-scale fermentation and purification; MS and $^{1}$H and $^{13}$C NMR (Supplementary Figs. 70, 71) data analyses of **9** (Supplementary Table 2) allowed us to determine the compound to be *N*-(1, 1-dimethyl-1-allyl)-tryptophan (Fig. 3c). This result demonstrated the necessity of *ilaS* in the construction of the ilamycin core, and also suggested that the Trp is first *N*-prenylated before being loaded onto the ilamycin biosynthetic assembly line. The *ila* cluster indeed contains an aromatic prenyltransferase, IlaO, that displays high identity to CymD in cyclomarin biosynthesis[22]. We inactivated *ilaO* and found that the Δ*ilaO* mutant completely lost the ability to generate both the ilamycins and fragment **9** (Fig. 2, trace iii). Consequently, IlaO appears to catalyze the reverse prenylation of Trp during a pre-tailoring step during ilamycin biosynthesis (Fig. 3c). The *ilaG* gene encoding a MbtH-like protein was also inactivated, the Δ*ilaG* mutant almost completely lost the ability to produce ilamycins (Fig. 2, trace iv), demonstrating the importance of IlaG in ilamycin biosynthesis and that its role is consistent with that of similar proteins present in other biosynthetic pathways[23–27]. We further narrowed the *ila* gene cluster from *ilaA* to *ilaT*, according to the analysis of HPLC results of ten mutants (Δ*orf(-2)* to Δ*orf(-6)* and Δ*orf(+1)* to Δ*orf(+5)*, Supplementary Fig. 27), which produce as much ilamycins as the wild-type producer.

**Biosynthesis of the L-AHA unit**. The ilamycins contain a L-AHA building block whose biosynthetic origin is unclear; the L-AHA structural element is unique and has not been observed in any other natural products. In our examination of the *ila* gene cluster, we observed a 4835 aa PKS, IlaE, consisting of three modules comprising 13 domains (Fig. 3e). In contrast to canonical type I PKSs, IlaE possesses three features. An AT-like (AT$_1$-L) domain and a canonical AT (AT$_2$) domain are assignable to IlaE, but the conversed active site residues S and R in the GXSXGE…R motif are mutated (Supplementary Fig. 31) and the malonyl-CoA specificity motif (HAFH) are present only in the later canonical AT (AT$_2$) domain[28], suggesting that the later canonical AT$_2$ domain maybe used iteratively to synthesize the C6 chain. Two keto synthases (KSs), KS$_2$ and KS$_3$, along with a small portion of the linker regions before and after KS$_2$ (*ca* 1500 bp in size) has completely identical DNA sequences. The giant PKS lacks a TE domain for releasing the polyketide chain. To investigate the exact role of IlaE in ilamycin biosynthesis, a Δ*ilaE*

mutant was constructed. The production of ilamycins was completely abolished in the Δ*ilaE* mutant (Fig. 2, trace v), suggesting that IlaE is necessary for ilamycin biosynthesis and might be involved in the biosynthesis of the unique L-AHA structural unit.

To confirm that the unique L-AHA unit found in the skeleton of ilamycins is indeed biosynthesized by IlaE, and to explore the biosynthetic origin of this unit, feeding experiments using [1-$^{13}$C], [2-$^{13}$C], and [1, 2-$^{13}$C]-labeled sodium acetate were conducted with *S. atratus* SCSIO ZH16, and the representative product, ilamycin B$_2$, was purified. Inspection of the $^{13}$C NMR spectra (Supplementary Fig. 44) of the $^{13}$C-labled ilamycin B$_2$ revealed that: (i) feeding with [1-$^{13}$C] acetate led to C$_{17}$, C$_{19}$, and C$_{21}$ enrichment in ilamycin B$_2$, (ii) feeding with [2-$^{13}$C] acetate led to C$_{18}$, C$_{20}$, and C$_{22}$ enrichment in ilamycin B$_2$, and (iii) feeding with [1, 2-$^{13}$C] acetate led to enrichment of C$_{17}$–C$_{22}$ in ilamycin B$_2$, and each of the C$_{17}$/C$_{18}$, C$_{19}$/C$_{20}$, and C$_{21}$/C$_{22}$ pairs appeared as coupled doublets ($^{1}J_{CC}$ = 25.2 Hz). These data convincingly demonstrate that acetate is the direct precursor of this unique C6 structural element, and that the L-AHA unit is assembled by IlaE, a type I PKS with clearly unusual features. IlaE does not contain a TE domain for the release of the C6 polyketide chain; transformation of the C6 polyketide chain to the L-AHA unit requires further amination at the α-position. Nearby *ilaE*, we found four genes that might be involved in these α-amination tailoring steps: *ilaC* encoding a hydrolase, *ilaD* encoding a cytochrome P450 monooxygenase, *ilaF* encoding a type II thioesterase, and *ilaH* encoding an aminotransferase. To probe if these four genes are involved in L-AHA biosynthesis, we individually inactivated each one to yield four mutants: Δ*ilaC*, Δ*ilaD*, Δ*ilaF*, and Δ*ilaH*. HPLC analysis of the fermentation extracts showed that the Δ*ilaD* strain failed to produce ilamycins (Fig. 2, trace vi); the Δ*ilaF* strain produced significantly lowered titers of ilamycins (Fig. 2, trace vii); and the titers of ilamycins from the Δ*ilaC* (Fig. 2, trace viii) and Δ*ilaH* (Fig. 2, trace ix) mutants were only one-half and two-thirds as great as the titers from wild-type producer, respectively. To further test if IlaD indeed takes part in the biosynthesis of the L-AHA unit and to determine the biosynthetic timing of L-AHA installation, as well as exclude the polar effect of *ilaE* knockout on the expression of the down stream genes, synthesized L(D)-AHA and 4-hexenoic acid (4-HA) were fed to the Δ*ilaD* mutant. Additionally, synthesized L(D)-AHA, 2,4-hexadienoic acid (2,4-HDA), and 4-HA were supplied to the Δ*ilaE* mutant strain. For each feeding experiment, synthetic precursors were added (individually) to a final reaction concentration of 0.5 mM in precursor. The fed mutant strains were all cultivated for 7 days alongside the wild-type strain as a control. HPLC analyses of the fermentation extracts revealed that ilamycins production was restored in both the Δ*ilaD* and Δ*ilaE* mutants when supplied with L(D)-AHA (Fig. 2, traces x and xii). However, neither mutant strain produced ilamycins when supplied with 4-HA to the Δ*ilaD* and Δ*ilaE*, as well as feeding 2,4-HDA to the Δ*ilaE* mutant also failed to restore ilamycin production (Fig. 2, traces xi, xiii, xiv).

Based on the above feeding results and combined with the gene knockout results using *ilaDEH*, we propose that IlaD performs an α-oxidation to form α-keto intermediate 10, and that the sole aminotransferase within the cluster, IlaH, is responsible for transamination of 10 to form the L-AHA unit. The oxidation of the C6 polyketide chain by IlaD and subsequent transamination at the α-position likely occurs while the substrate is tethered to the ACP$_3$ of IlaE (Fig. 3e). However, the chemical complementation experiment with AHA demonstrates that the L-AHA unit is released from the PKS before being loaded onto the NRPS assembly line to form ilamycins. Two enzymes likely able to hydrolyze and release the L-AHA unit from IlaE are the hydrolase IlaC and the type II TE IlaF. Both of these kinds of enzymes have

been shown to have similar functions in other antibiotic biosyntheses[29–31]. Although, the ΔilaC and ΔilaH mutants still produced ilamycins, other similar enzymes within the SCSIO ZH16 genome may complement their functions.

**Biosynthesis of the L-3-nitrotyrosine unit**. Nitro-containing natural products possess diverse structures and remarkable biological activities. However, the biosynthesis of nitro groups in natural products is poorly elucidated, and only a limited number of the enzymes involved in the formation of aromatic nitro groups have been characterized[32]. The mechanism for tyrosine nitration remains an unsolved mystery. Nevertheless, a growing number of studies reveal that the presence of 3′-nitro-tyrosine and 3′-nitro-tyrosine-modified proteins are closely related to or serve as a biomarker for many diseases, including atherosclerosis, Parkinson's disease, cardiomyocyte disease, respiratory disease, Alzheimer's disease, and various kinds of cancers and infectious diseases[33–37]. The presence of an L-3-nitrotyrosine unit in the ilamycins provides an excellent opportunity to elucidate the biosynthetic mechanism/s leading to this rare unit. In the biosynthetic gene cluster of ilamycins, ilaM and ilaN encode a nitric oxide synthase and a cytochrome P450 oxygenase, respectively. Silico analysis revealed that IlaM shared the same active sites with other nitric oxide synthase originate from the human or murine, but clustered in different clades[38] (Supplementary Fig. 32). IlaM also shows sequence homology to TxtD (52% identity) from S. turgidiscabies car8, which has been proposed to generate nitric oxide from L-Arg for further use in L-Trp 4-nitration catalyzed by the cytochrome P450, TxtE, in the thaxtomin pathway[39] (Supplementary Fig. 33). Inspired by this realization, we proposed that the formation of the L-3-nitrotyrosine might be similarly catalyzed by IlaM and the downstream IlaN.

To validate this hypothesis, the ilaM and ilaN genes were each inactivated using the aforementioned method. HPLC analyses of the fermentation extracts of the resultant mutant strains revealed that both the ΔilaM and ΔilaN mutants failed to produce ilamycins or their analogs (Fig. 2, traces xv and xvi). These results, along with the bioinformatics analysis, suggest that IlaM and IlaN might be responsible for pre-tailoring L-Tyr to L-3-nitrotyrosine since no evidence of de-nitro ilamycin analogs could be found.

In order to confirm that the nitration of L-Tyr is a pre-tailoring process in the biosynthesis of ilamycins and that the nitration is catalyzed by IlaM and IlaN, L-3-nitrotyrosine (0.5 mM) was individually fed to cultures of the ΔilaM and ΔilaN strains and each was cultured for 7 days. The production of ilamycins was restored in each of the two mutants (Fig. 2, traces xviii and xix). These results clearly establish that the nitration of tyrosine occurs before L-3-nitrotyrosine is loaded onto the ilamycin NRPS assembly line and not after NRPS elongation in ilamycin biosynthesis (Fig. 3d). Moreover, these data suggest that mutations to ilaM and ilaN might enable a mutasynthetic approach to ilamycin analogs with different bioactivities and that could be produced by fermentation doping with different precursors. Alternatively, IlaM and IlaN could be used to de novo biosynthesize the L-3-nitrotyrosine building block, which can be a valuable and promising strategy for drug discovery since the presence of nitro groups in antibiotics such as aureothin, delamanid, and other nitro-heterocyclic compounds is known to alter bioactivities[40–42].

**Post-tailoring steps in ilamycin biosynthesis**. We next elucidated the post-tailoring steps in the biosynthesis of ilamycins. Among all of the ilamycin analogs, Ilamycin D (**5**) is the most highly oxidized; a terminal methyl group in the N-methyl Leu unit is oxidized to its COOH moiety and the double bond of the isopentene is oxidized to an epoxy group (Fig. 3f). The p450 enzymes with diverse oxidation functions maybe the better candidates to fulfill these roles[43]. In the ila gene cluster, there is a cytochrome P450 monooxygenase IlaL in the middle of the cluster and another cytochrome P450 monooxygenase IlaR adjacent to the giant NRPS protein IlaS. Both of these p450s may be responsible for these post-tailoring oxidation steps. Similarly, the ilaL and ilaR genes were each inactivated and the fermentation extracts of the mutant strains were analyzed by HPLC (Fig. 2, traces xvii and xx).

$^1$H, $^{13}$C NMR, and HRMS data of the compounds isolated from fermentations of the ΔilaL mutant demonstrated that they were identical to ilamycin B$_1$ (**1**) and ilamycin B$_2$ (**2**). Since the ΔilaL mutant was found to generate only compounds 1 and 2, the latter of which has an epoxy group, we propose that IlaL is not essential for epoxide biosynthesis. Rather, deductive reasoning and by process of elimination, it appears that IlaL plays a central role in the net 6-electron oxidation of L-Leu to install the lone carboxylic acid observed in certain ilamycins. We envision that COOH installation at the hands of IlaL proceeds through an aldehyde intermediate (Fig. 3f).

HPLC analysis revealed that the ΔilaR mutant accumulated four products, **1** and **6–8** (Fig. 2, trace xx), and LC-MS analysis of the metabolites revealed that the latter two adducts might be other ilamycin analogous. To elucidate the exact structures of these metabolites, cultures of the ΔilaR mutant were scaled-up in a 16L fermentation, and the four compounds, **1** and **6–8**, were subsequently isolated and purified. Compounds **1** and **6** isolated from the ΔilaR mutant were identified as ilamycin B$_1$ and ilamycin E$_1$, respectively, on the basis of HRESIMS and $^1$H and $^{13}$C NMR spectroscopic data comparisons. The molecular formula of compound **7** (designated ilamycin E$_2$) was established to be $C_{54}H_{75}N_9O_{11}$ by HRESIMS. The planar structure of **7** coincided with that of ilamycin E$_1$ (**6**), which was elucidated in the same fashion. Comparisons of NMR data (Supplementary Figs. 60–64) further showed that ilamycins E$_1$ and E$_2$ (**6** and **7**) have the same stereochemistries at the γ- and δ-carbons ($C_{32}$ and $C_{33}$) of the cyclic hemiaminals containing each N-methyl leucine unit as seen in ilamycins C$_1$ and C$_2$ (**3** and **4**), respectively. The molecular formula of compound **8** (designated ilamycin F) was established to $C_{54}H_{75}N_9O_{12}$ by HRESIMS, 16 mass units less than that of ilamycin D (**5**). The $^1$H and $^{13}$C NMR spectroscopic data (Supplementary Table 2 and Supplementary Figs. 65, 66) of **8** were very similar to that of ilamycin D, except that the signals ascribed to the epoxy group in ilamycin D were missing in spectra of **8**. Conversely, additional signals at $\delta_C$ 145.5, $\delta_H$ 6.18 (CH-13) and $\delta_C$ 114.1, $\delta_H$ 5.25, 5.21 (CH$_2$-14) in the spectrum of **8** were clearly present indicating the presence of a terminal double bond. The COSY, HMQC, and HMBC correlations (Supplementary Figs. 67–69) confirmed the planar structure of compound **8**. Analysis of the X-ray diffraction data for **8** revealed that the γ-C ($C_{32}$) stereochemical configuration in the 2-amino-4-methylpentanedioic acid residue is S (Supplementary Table 1; Supplementary Fig. 4 and Supplementary Data 4, 5). The structural elucidation of these four products in the ilaR mutant fermentations allowed us to assign IlaR as an isopentene epoxidase (Fig. 3f).

An interesting feature we identified during HPLC studies is that compounds **7** and **6** epimerize/interconvert in aqueous solvents (e.g., MeOH or CH$_3$CN); this interconversion also was noted under neat conditions following periods of long-term storage. The ratio of compound **7** to **6** in MeOH was found to be ca 6:4 at equilibrium and compound **6** is reasonably stable in non-aqueous solvents. At room temperature and under acidic aqueous condition (15% CH$_3$CN in H$_2$O containing 0.1% acetic acid) or in MeOH, purified **6** was found to slowly convert to **7**, and purified **7** was found to slowly convert to **6** (Supplementary

Fig. 34A); similar chemistry was noted under neat conditions but only after prolonged periods. Similarly, such conditions were found to facilitate more rapid conversion of compound **4** into **3**; at equilibrium we found the ratio of 4:3 to be *ca* 5:5. The possibility interchange of these two pairs of compounds was postulated (Supplementary Fig. 34B). Interestingly, a similar reaction able to take place under neat (solvent and catalyst-free) conditions to form bicyclic hemiaminals using a pair of substrates containing an aldehyde group and *N*-substituted amide group has been noted[44]. It is indeed interesting to consider that cyclic hemiaminal installation in ilamycins C$_1$, C$_2$, E$_1$, and E$_2$ is likely reflective of a transient C$_{33}$ aldehyde and that the αOH orientation is preferred; presumably this is driven to some extent by the fixed stereochemical orientations at C$_{24}$ and C$_{32}$ in each respective case (Fig. 3f). By virtue of this interconversion chemistry it is significant to note that assays carried out with these agents likely involved the use of isomeric mixtures of **6/7** and **3/4** and no one specific isomer.

**Antimycobacterial and cytotoxic activities of ilamycins.** Having never been reported previously, we assessed the cytotoxicity of **1**–**8** using five human tumor cell lines and two normal human cell lines. The results demonstrated that compounds **3/4**, **6/7** exhibited cytotoxic activities against HeLa, HepG2, and A549 cell lines with IC$_{50}$ values in the range 3.2–6.2 µM (Supplementary Table 6). Additionally, the cytotoxic ilamycins generally displayed a 3–5-fold reduced activity against two normal cell lines; these agents showed a clear preference for harming cancerous cell lines.

Finally, we systematically evaluated antimicrobial activities of compounds **1**–**8** using a panel of six Gram-positive and Gram-negative bacteria, and two mycobacteria, including *M. smegmatis* MC$^2$ 155 and *M. tuberculosis* H37Rv. The ilamycins failed to show antibacterial activities (MICs > 121 µM) against the first six bacteria (Supplementary Table 7). However, selective activities against the mycobacteria were, in some cases, quite prominent. Most notably, ilamycins E$_1$/E$_2$ (**6/7**) showed the strongest inhibitory activity against *M. tuberculosis* H37Rv with an MIC value of 9.8 nM, which was 30-fold superior to that of positive control, rifampin (Table 1). More intriguingly, **6** and **7** (likely as a mixture) bear a therapeutic activity/toxicity index of 400–1500, indicating **6/7** hold great promise for anti-tubercular drug discovery. Furthermore, ilamycins D (**5**) and F (**8**) also showed strong anti-tubercular activities against *M. tuberculosis* H37Rv with an MIC value of 1.2 µM.

In summary, eight ilamycin congeners (**1**–**8**) with the determined absolute stereochemistries were isolated from the deep South China Sea-derived *S. atratus* SCSIO ZH16 and its genetic engineered mutants. Most notably, the compounds (**6/7**, **8**) showed especially active against *M. tuberculosis* H37Rv with MICs of 9.8 nM and 1.2 µM, respectively, can be produced in high titers (*ca* 13.5 mg L$^{-1}$, 12.5 mg L$^{-1}$) by a same genetic engineered mutant (ΔilaR). They will hold a great promise as the lead drugs for anti-TB agents. Moreover, the biosynthetic mechanisms of ilamycins characterized with three pre-tailoring steps (the prenylation of tryptophan, the biosynthesis of L-AHA, and the nitration of tyrosine) and two post-tailoring steps (the carboxylation of the C$_{33}$ methyl group in the Leu residue and the epoxidation of prenyl-L-tryptophan) were probed by gene inactivation studies, chemical precursor complementation, and stable isotope-labeled precursor feeding experiments. These efforts have led not only to the elucidation of the biosynthetic timing and mechanisms driving ilamycins production but have also unveiled three potent anti-TB agents as drug candidates.

## Methods

**General materials and experimental procedures.** NMR spectra were obtained with an AVANCE-500 spectrometer (Bruker). CC was performed using silica gel (100–200 mesh; Jiangyou). Medium-pressure liquid chromatography was performed using a CHEETAH 100 automatic flash chromatography system (Bonna-Agela) with an ODS-A flash column (S-50 µm, 12 nm; 100 × 20 mm, YMC). Semi-preparative HPLC was performed with two 210 solvent delivery modules equipped with a 335 photodiode array detector (Hitachi), using an ODS-A column (250 × 10 mm, 5 µm, YMC). Low-resolution and high-resolution mass spectra were obtained on an Amazon SL ion trap instrument and a Maxis quadrupole-time-of-flight mass spectrometer (Bruker), respectively. Single-crystal data were collected on an Xcalibur Onyx Nova diffractometer (Oxford) using Cu Kα radiation or a Gemini S Ultra X-ray diffraction system (Rigaku, Oxford).

Antibiotics were purchased from Sangon Biotech Co., Ltd. (Shanghai, China) and Fisher-Scientific (Waltham, MA, USA), respectively. The isotope-labeled compounds were purchased from Cambridge Isotope Laboratories (Tewksbury, MA, USA). L-3-nitrotyrosine was purchased from Alfa Aesar (Shanghai, China). Synthetic L(D)-AHA, 2, 4-HDA, and 4-HA were purchased from Ningbo Kangbei Biochemical Co., Ltd. (Ningbo, China). All chemicals and solvents were of analytical or chromatographic grade.

**Bacterial strains and plasmids.** Strain SCSIO ZH16 was isolated from a sediment sample collected from the South China Sea (120°0.250′E, 20°22.971′N) at a depth of 3536 m. Phylogenetic analysis based on the nearly complete 16S rRNA gene sequence indicated that strain SCSIO ZH16 belongs to the genus *Streptomyces*. The highest 16S rRNA gene sequence similarity value was 100% between strain SCSIO ZH16 and *Streptomyces atratus* PY-1 (KJ627770). The 16S rRNA gene sequence has been deposited in GenBank under accession number KT9708. This strain was deposited in the China General Microbiological Culture Collection Center, Institute of Microbiology, Chinese Academy of Sciences (Beijing, China) as *Streptomyces atratus* SCSIO ZH16. Strains and plasmids used and generated in this study are listed in Supplementary Table 3.

**Culture conditions and DNA manipulations.** *S. atratus* SCSIO ZH16 was cultured on modified ISP$_2$ (0.4% yeast extract, 1.0% malt extract, 0.4% glucose, and 3.0% crude sea salt) plates with additional 20 mM MgSO$_4$ at 30 °C. For the isolation of ilamycins or their analogs, seed cultures of *S. atratus* SCSIO ZH16 were grown in Am2ab medium for 60 h and then inoculated into Am3 production medium (0.5% soybean meal, 1.5% bacterial peptone, 1.5% soluble starch, 1.5% glycerol, 0.2% CaCO$_3$, and 3% sea salt, pH 7.2–7.4) at a ratio of 1:10 before being cultured for another 7 days at 30 °C and 200 rpm. All DNA isolation and manipulation procedures in *Escherichia coli* and *Streptomyces* were performed according to standard procedures or the manufacturer's protocol. Primers were synthesized by Sangon Biotech Co., Ltd. (Shanghai, China). DNA sequencing was performed at IGE Biotech Co., Ltd. (Guangzhou, China). Restriction enzymes and DNA ligase were purchased from Takara Biotechnology Co., Ltd. (Dalian, China). Plasmid, gel extraction, and cycle-pure kits were acquired from Omega Bio-Tek Inc. (GA, USA). PCR amplifications were carried out using either EasyTaq or high-fidelity polymerase purchased from Transgene Biotech Co., Ltd. (Beijing, China).

**Purification of compounds 1–9.** For the isolation of compounds **1**–**6**, the strain *S. atratus* SCSIO ZH16 was cultivated on an 18 L of Am3 medium using the methods mentioned above. After 8 days cultivation, the whole culture medium was centrifuged to separate the liquid broth from the solid cell mass. The liquid broth and the solid cell mass were then extracted with an equal volume of butanone and a double volume of acetone, respectively. The organic layer was dried under vacuum, and the two parts of the extracts were combined.

The extract was applied to silica gel CC using a gradient elution of CHCl$_3$/MeOH (100:0, 98:2, 96:4, 94:6, 92:8, 9:1, 8:2, and 5:5) to give eight fractions (Fr. A1–A8). Fr. A2 and A3 were combined and purified by silica gel CC, eluting with gradient ratios of petroleum ether/ethyl acetate (100:0, 8:2, 6:4, 4:6, 2:8, and 0:100), ethyl acetate/MeOH (95:5), and CHCl$_3$/MeOH (9:1) to give eight fractions (Fr.

**Table 1 Antimycobacterial activities of 1–8**

| | M. smegmatis MC$^2$ 155 | M. tuberculosis H37Rv |
|---|---|---|
| **1** | >126.5 | 98.9 |
| **2** | >124.5 | 2.4 |
| **3/4** | 0.12 | 9.6 |
| **5** | 30.3 | 1.2 |
| **6/7** | 30.7 | 0.0098 |
| **8** | 30.7 | 1.2 |
| [a]Kan/[b]Rif | 1.7[a] | 0.3[b] |

*Note:* Results expressed as MICs (µM)
The anti-TB activities of compounds **1-8** were performed in triplicate, *n* = 3
[a]Kanamycin
[b]Rifampin

B1–B8). Fr. B5–B8 were each separated using semi-preparative HPLC with an ODS column, eluting with $CH_3CN/H_2O$ (40:60–100:0 over 30 min, 10 mL min$^{-1}$) to yield ilamycin B$_1$ (1, 50 mg), ilamycin E$_1$ (6, 18 mg), and ilamycin B$_2$ (2, 22 mg); Fr. A4 and Fr. A5 were purified by silica gel CC, eluting with gradient ratios of petroleum ether/ethyl acetate (100:0, 8:2, 6:4, 4:6, 2:8, and 0:100), ethyl acetate/ MeOH (95:5), and CHCl$_3$/MeOH (9:1) to give eight fractions (Fr. C1–C8). Fr. C7 and Fr. C8 were each separated using semi-preparative HPLC with an ODS column, eluting with $CH_3CN/H_2O$ (30:70–100:0 over 30 min, 10 mL min$^{-1}$) to yield ilamycin C$_1$ (3, 28 mg), ilamycin C$_2$ (4, 33 mg), and ilamycin D (5, 78 mg).

Based on HPLC analysis results of the mutant cultures, we selected $\Delta ilaL$, $\Delta ilaR$, and $\Delta ilaS$ mutants for large scale fermentation using the aforementioned methods. Finally, 88 mg of 1, and 152 mg of 2 were isolated from a 13.5 L fermentation of the $\Delta ilaL$ mutant, 130.5 mg of 6, 85.6 mg of 7, 240.3 mg of 8, and 150.5 mg of 1 were isolated from a 16 L fermentation of the $\Delta ilaR$ mutant, and 5.6 mg of 9 was isolated from a 13.5 L fermentation of the $\Delta ilaS$ mutant using isolation procedures similar to those described above.

Ilamycins E$_1$ (6): Yellow powder; $^1H$ and $^{13}C$ NMR data were summarized in Supplementary Table 2, 2D (COSY, HSQC, and HMBC) NMR spectra, see Supplementary Figs. 57–59; HRESIMS $m/z$ 1026.5669 [M+H]$^+$ (calculated for C$_{54}$H$_{76}$N$_9$O$_{11}^+$, 1026.5659), see Supplementary Fig. 38.

Ilamycins E$_2$ (7): Yellow powder; $^1H$ and $^{13}C$ NMR data were summarized in Supplementary Table 2, 2D (COSY, HSQC, and HMBC) NMR spectra, see Supplementary Figs. 62–64; HRESIMS $m/z$ 1026.5673 [M+H]$^+$ (calculated for C$_{54}$H$_{76}$N$_9$O$_{11}^+$, 1026.5659), see Supplementary Fig. 39.

Ilamycins F (8): Yellow powder; $^1H$ and $^{13}C$ NMR data were summarized in Supplementary Table 2, 2D (COSY, HSQC, and HMBC) NMR spectra, see Supplementary Figs. 67–69; HRESIMS $m/z$ 1042.5615 [M+H]$^+$ (calculated for C$_{54}$H$_{76}$N$_9$O$_{12}^+$, 1042.5608), see Supplementary Fig. 40.

N-(1, 1-dimethyl-1-allyl)-tryptophan (9): White powder; $^1H$ and $^{13}C$ NMR data were summarized in Supplementary Table 2, $^1H$ and $^{13}C$ NMR spectra, see Supplementary Figs. 70 and 71; ESIMS $m/z$ 273.2 [M+H]$^+$, see Supplementary Fig. 41.

**X-ray crystallographic analysis of compounds 2, 4, 5, and 8.** Four yellow block crystals (2, 4, 5, and 8) were obtained from MeOH, MeOH-CHCl$_3$ (9:1), MeOH-EtOH (1:1), and MeOH-EtOH (4:1), respectively. The crystal data of 2 were recorded on an Oxford Gemini S Ultra single-crystal diffractometer with enhanced Mo Kα radiation ($\lambda = 0.71073$ Å). The structure was solved by direct method ShelXL and refined using full-matrix least-squares difference Fourier techniques[45], [46]. The crystal data of 4, 5, and 8 were recorded on an Oxford Xcalibur single-crystal diffractometer with enhanced ultra Cu Kα radiation ($\lambda = 1.54184$ Å). The structures were solved by direct methods ShelXL and refined using full-matrix least-squares difference Fourier techniques[47]. The crystal parameters of these four crystals were listed in Supplementary Table 1. The perspective views of the X-ray crystal structures of 2, 4, 5, and 8 were shown in Supplementary Figs. 1–4. Crystallographic data for 2, 4, 5, and 8 have been deposited in the Cambridge Crystallographic Data Centre with the deposition number CDCC1524774, CDCC1524775, CDCC1524776, and CDCC1524777, respectively (Supplementary Data 1–5). A copy of the data can be obtained, free of charge, on application to the Director, CCDC, 12 Union Road, Cambridge CB2 1EZ, UK (fax: +44(0)-1233-336033 or e-mail: deposit@ccdc.cam.ac.uk).

**Genome sequencing and bioinformatics analysis.** The genomic DNA of S. atratus SCSIO ZH16 was sequenced using a combination of second-generation 454 and Illumina HiSeq 4000 sequencing technologies and third-generation PacBio sequencing technology at Shenzhen BGI Diagnosis Technology Co., Ltd. and Shanghai Biozeron Co., Ltd. The assembled genome sequence was subjected to analysis for secondary metabolite biosynthetic gene clusters by the online anti-SMASH software (http://antismashsecondarymetabolites.org/)[18]. ORFs were analyzed using the online FramePlot 4.0 beta software (http://nocardia.nih.go.jp/fp4/), and their functional predictions were obtained using an online BLAST program (http://blast.ncbi.nlm.nih.gov/). The NRPS and PKS domains were predicted using the PKS/NRPS analysis website (http://nrps.igs.umaryland.edu/nrps/)[48]. The gene cluster for ilamycins (ila) was deposited in GenBank under the accession number KY173348. The deduced orf functions in ila biosynthetic gene cluster were shown in Supplementary Table 4.

**Genomic library construction and screening.** The S. atratus SCSIO ZH16 genomic cosmid library was constructed using SuperCos 1, according to the manufacturer's protocol. About 2600 clones were picked into 96-well plates and stored at −80 °C. Four pairs of primers associated with the ilaE, ilaL, ilaR, and ilaS (Supplementary Table 5) were designed and used to screen the genomic cosmid library using PCR.

**Gene inactivation experiments.** The λ-Red-mediated gene replacements were performed following standard procedures. The gene inactivation experiments in S. atratus SCSIO ZH16 were carried out as previously reported. Briefly, apra-mycin resistance gene cassettes aac(3)IV-oriT with 39 bp homologous to each side of the gene to be inactivated were amplified by PCR using the primers listed in

Supplementary Table 5. Each of the apramycin resistance gene cassettes was then introduced into E. coli BW25113/pIJ790/201E, BW25113/pIJ790/23D, or BW25113/pIJ790/47H by electroporation depending on the location of the gene to be inactivated. Correct mutants were verified by PCR amplification and restriction enzyme digestion of the mutant cosmid. The correctly mutated cosmids were then introduced into E. coli ET12567/pUZ8002 for conjugation with S. atratus SCSIO ZH16. The conjugation processes were carried out as described previously[20], and double crossover mutant strains possessing the kanamycin$^S$ and apramycin$^R$ phenotypes were confirmed by PCR using primers listed in Supplementary Table 5. Finally, 22 S. atratus SCSIO ZH16 genetic mutants ($\Delta ilaCDEFGH$, $\Delta ilaLMNO$, $\Delta ilaRS$, $\Delta orf(+1)$ to $\Delta orf(+5)$, and $\Delta orf(-2)$ to $\Delta orf(-6)$) were successfully constructed. The PCR verification of the mutants were shown in Supplementary Figs. 5–26. The HPLC analysis of the metabolite profiles of 10 mutants ($\Delta orf(+1)$ to $\Delta orf(+5)$, and $\Delta orf(-2)$ to $\Delta orf(-6)$) showed that they were not necessary for the biosynthesis of ilamycins as they did not produce an obvious influence on ilamycin generation (Supplementary Fig. 27).

**Silico analysis of the domains in IlaE and IlaM.** To investigate the possible mechanisms underlying the unique modules of IlaE, multiple sequence alignments were carried out for each type of domain. The sequence alignments results revealed that the conserved motifs in KS$_1$ (Supplementary Fig. 28), KR$_2$, KR$_3$ (Supplementary Fig. 29), and ER$_3$ (Supplementary Fig. 30) were different from the identified conserved ones[49–56]. The substrates specificities of AT domain were predicted based on the conserved sequence of GHSIGE…R…HAFH (Supplementary Fig. 31). Silico analysis of indicted that IlaM shared the same active sites with other nitric oxide synthase originate from the human or murine, but clustered in different clades (Supplementary Fig. 32). IlaM also shows sequence homology to TxtD (52% identity) from Streptomyces turgidiscabies car8 (Supplementary Fig. 33), which has been proposed to generate nitric oxide from L-Arg for further use in L-Trp 4-nitration catalyzed by the cytochrome P450, TxtE, in the thaxtomin pathway[37, 38].

**Metabolite analysis of wild-type and mutant strains.** To analyze the metabolites of each mutant, the mutant strains were inoculated in a 250-mL flask filled with 50-mL Am2ab medium and grown at 30 °C on a rotary shaker at 200 rpm for 7 days with the wild-type as a control. The fermentation was extracted with an equal volume of butanone and processed using the aforementioned method. The dried extracts of the fermentation products were re-suspended in 1-mL methanol and were centrifuged for 10 min at 14000 rpm before HPLC analysis. HPLC analysis was carried out using a reversed phase column SB-C18, 5 μm, 4.6 × 150 mm (Agilent) with UV detection at 210, 285, and 352 nm under the following program: solvent system (solvent A, 15% acetonitrile in water supplemented with 0.1% acetic acid; solvent B, 85% acetonitrile in water supplemented with 0.1% acetic acid); 20% B to 80% B (linear gradient, 0–20 min), 80% B to 100% B (linear gradient, 20–21.5 min), 100% B (21.5–27.0 min), 100% B to 0% B (27.0–27.1 min), 0% B (27.1–30.0 min); flow rate was set at 1 mL min$^{-1}$.

**Isotopic labeling experiments.** To elucidate the biosynthetic origin of the L-AHA unit, isotopic labeling experiments were carried out with cultures of the S. atratus SCSIO ZH16. Three $^{13}C$-labled compounds, [1-$^{13}C$] sodium acetate, [2-$^{13}C$] sodium acetate, and [1, 2-$^{13}C$] sodium acetate, were used for the feeding experiments. The $^{13}C$-labeled agent was dissolved in ddH$_2$O at a concentration of 500 mg mL$^{-1}$ as stock solution. The stock solution was sterilized via filtration, and then individually supplied into 1 L production cultures (Am3 medium) at 60, 72, and 84 h with the volume of 0.5, 1, and 0.5 mL, respectively. After cultivation at 220 rpm and 30 °C for 7 days, the cultures were harvested and extracted. The representative product, $^{13}C$-labeled ilamycin B$_2$, was purified from the organic extract.

**Precursor feeding experiments.** A portion of mycelium and spores (1 cm$^2$) of the mutant strain was inoculated in a 250-mL flask with 50 mL of Am3 medium as described above at 30 °C and 200 rpm for 60 h. Synthesized L(D)-AHA, 2,4-HDA, and 4-HA were dissolved in acidified water, neutralized with 1 N NaOH to pH 7.0, and sterilized via filtration. The sterilized L(D)-AHA and 4-HA were supplied into the culture of the $\Delta ilaD$ mutant, and the L(D)-AHA, 2,4-HDA, and 4-HA were individually fed to cultures of $\Delta ilaE$ mutant to achieve a final concentration of 0.5 mM. Following incubation at 30 °C and 200 rpm for 7 days, the cultures were each harvested and extracted with butanone, evaporated to dryness, and dissolved in MeOH for HPLC analysis (Fig. 2, x–xv). The feeding procedures of 3-nitro-L-tyrosine to $\Delta ilaM$ or $\Delta ilaN$ mutants were similarly conducted. The 3-nitro-L-tyrosine precursor was dissolved in acidified water, neutralized with 1 N NaOH to pH 7.0 and sterilized via filtration, and was then individually supplemented into the cultures of $\Delta ilaM$ or $\Delta ilaN$ mutants (Fig. 2, xviii and xix).

**Cytotoxicity assays.** Compounds 1–8 were evaluated for cytotoxicity using HeLa, HepG2, A549, CNE2, and MCF7 cell lines using a previously reported MTT method[57]. The L02 and Huvec-12 cell lines were used as normal cell line controls. All these cell lines were obtained from American Type Culture Collection (ATCC) and cultured according to ATCC recommendations. Cell lines were checked for mycoplasma and profiled via short tandem repeat profiling to confirm their

identity by the supplier. All experiments were performed in triplicate with dox-orubicin and cis-platinum as control agents.

**Antibacterial assays.** To determine the antibacterial activities of these eight compounds (1–8), a preliminary screening of their antibacterial activities against a panel of Gram-positive and Gram-negative bacteria[6–8], M. smegmatis MC[2] 155 and M. tuberculosis H37Rv, was conducted using a broth dilution method[58, 59]. Compounds 1–8 were dissolved in dimethylsulfoxide (DMSO) to give 3200 μg mL$^{-1}$ stock solutions. The stock solutions were then serially diluted to concentrations of 0.0625–128 μg mL$^{-1}$ with MH broth or 7H9 broth (added with 0.2% glycerol, 0.05% Tween 80). DMSO in MH broth or 7H9 broth was used as a negative control, and broth-containing bacteria was used as positive control. All experiments were performed in triplicate with kanamycin and rifampicin as control agents.

**Data availability.** Sequence data that support the findings of this study has been deposited in GenBank with accession codes KY173348 for ilamycin gene cluster and KT9708 for 16S rRNA gene sequence of S. atratus SCSIO ZH16. Deposition number of crystallographic data for 2, 4, 5, and 8 are CDCC1524774, CDCC1524775, CDCC1524776, and CDCC1524777, respectively. The authors declare that all other relevant data supporting the findings of this study are available within the article and its Supplementary Information files and from the corresponding author upon reasonable request.

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

## Acknowledgements

This work was supported in part by NSFC (31270134, 81425022, U1501223, 81572037), CAS (XDA11030403, YIPA-2013226, 154144KYSB20150045), Guangdong NSF (2016A030312014, 2016A030310123), the Pearl River S&T Nova Program of Guangzhou, China (2014J2200089) and Guangzhou Healthcare and Cooperative Innavation Major Project (201508020248, 201604020019). Additionally, we thank the analytical facility center (Ms. A. Sun, Dr. Z. Xiao, and Mr. C. Li) of the South China Sea Institute of Oceanology for recording MS and NMR data.

## Author contributions

J.J. and J.M. designed the research and wrote the paper. J.M. and Y.X. did the bioinformatics analysis. J.M., C.Z., and Y.Z. constructed all the engineered strains and mutants, as well as HPLC analysis. J.M. and Y.J. carried out the isotopic labeling and precursors feeding experiments. Y.J. and J.M. isolated all the compounds. J.J. and H.H. analyzed the NMR and X-ray diffraction data, and performed the structure determination of isolated compounds. Z.L. and T.Z. carried out antibacterial assays; J.Z. and H.Z. did cytotoxicity assays.

## Additional information

**Competing interests:** The authors declare no competing financial interests.

