## [Peer Review File · Nature Communications]

Reviewers' comments:

Reviewer #1 (Remarks to the Author):

In this interesting study, a family of compounds from marine bacterium *Streptomyces atratus* was analyzed, revealing this strain to be a producer of the known compounds known as ilamycins (rufomycins). Follow up structural analysis completed the determination of absolute stereochemistry for this family, identified and validated the gene cluster and facilitated the identification of some analogs in the series. In all 12 mutants and 5 chemical complementation experiments were performed for functional analysis and 10 additional replacements defined cluster boundaries.

Some proposals of the biosynthesis were generated based on gene replacement and chemical complementation studies. The nitration system is highly analogous to thaxtomin biosynthesis, and is as one would expect for this compound (at an activated position ortho to phenol). The polyketide synthase has some unusual features, especially a missing AT domain (see comment below). These studies, which complete the structure determination of a known series and inaugurate work into the biosynthesis are well executed, carefully described, and generally interpreted soundly (See comments below).

The genuinely surprising result from these studies is the observation that the analogs that accumulate because of disruption of an epoxidase (E1/E2) have specific nanomolar activity against tuberculosis, which are over 100-fold more active than closely related analogs, and substantially more potent than benchmark compounds. This is surprising because the electrophilic epoxides (C1/C2) are much less active than the terminal alkene E1/E2, and the aminal is only active in the presence of the terminal alkene. It appears from these data that both the aminal and the terminal alkene are essential. Given the clinical significance of tuberculosis, the potency of these newly elucidated compounds, and modest cytotoxicity a range of human cell lines (400 - 1900 fold less active in mammalian lines), these compounds could be important lead compounds in the development of new therapeutics. This result should be of interest to readers of Nature communications.

Comments:

It is hard to parse this sentence: "All the binding pocket sequences of the A domains in IlaS do not match their predicted amino acid substrates, indicative of the diversity of A domains and supporting the potential of their application in generating "engineered" natural products." Firstly, it is typical for automated predictions of of A-domain selectivity to yield ambiguous results. Secondly, that a homology model or neural network algorithm is unable to infer a prediction has no connection to biosynthetic plasticity or engineering potential

RE the missing AT domain, have the authors gone beyond Antismash for domain analysis. Antismash, which is unsupervised and overly reliant on Markov models, is not sufficient for this purpose. The presence/absence of a divergent AT domain should be ascertained by BLAST analysis as well. The presence/absence of a AT-like sequence between KS2/DH1 should be confirmed and commented on.

Error in figure 2e IlaE, should be a KSQ (superscript Q) in the domain. Probably a graphics software glitch.

Parag starting 212 needs some qualification. The conclusions are consistent with the chemical complementation data, but in absence of biochemical experiments, still preliminary. For instance, transport issue may affect uptake of precursors.

Parag 309. This is paragraph facile, it is completely expected that the aminals would interconvert as they are in equilibrium the aldehyde. Same phenomenon observed in anthramycin, for instance.

6/7 are epimers from simple amination equilibration hence are really one compound. This is typical and expected, for instance, see anthramycin.

It would improve the manuscript to have a little more understanding for the remarkable activity of 6/7. It is possible that the epoxide is unstable under assay conditions. It may open to the vicinal-diol via hydrolysis, resulting in a structural change that decreases activity. Have the authors checked the stability of the epoxide under bioassay conditions by HPLC/MS?

Reviewer #2 (Remarks to the Author):

Ma et al describes the identification of the ilamycin biosynthetic gene cluster from a *Streptomyces atratus* strain with rare L-3-nitro-tyrosine and L-2-amino-4-hexenoic acid (L-AHA) structural features. By a series of gene inactivation and feeding experiments with precursors including isotope-labeled precursors and subsequent structural elucidation of the accumulated intermediates, they clarified the biosynthesis steps and involved genes for L-3-nitrotyrosine and L-AHA, as well as the order of tryptophan prenylation and isoprene epoxidation. This work sounds solid and the manuscript is well written and can be easily followed. I recommend it for acceptance with minor revision.

Minor points

Lines 216 and 217

The authors concluded the release of L-AHA from the PKS before being loaded onto the NRPS assembly line, based on the feeding experiment with AHA.

Can we exclude the possibility that the observed results with AHA represent just an artificial effect?

Lines 237 and 258: Silica analysis?

Reviewer #3 (Remarks to the Author):

The unusual structure of ilamycins, containing rare L-2-amino-4-hexenoic acid (L-AHA) and L-3-nitro-tyrosine structural motifs makes them interesting starting points for biosynthetic studies. In addition, two ilamycin congeners show potent anti-tuberculosis activity (MIC values in the low nanomolar range), which makes them biomedically relevant as well.

The manuscript by Ma et al. is a comprehensive study of ilamycins, describing 1) the genome sequence of the producing bacterium (*Streptomyces atratus*); 2) the identification of the ilamycin biosynthetic gene cluster (BGC); followed by 3) the generation of 22 deletion mutants that allowed the authors to delineate the borders of the BGC and to probe the function of structural genes; 4) the absolute stereochemistry of ilamycin congeners; 5) insights into the biosynthesis of the unusual building blocks L-AHA and L-3-nitro-tyrosine; and 6) biological activity of ilamycins.

My recommendation is to publish the manuscript after minor revisions. Specific comments/recommendations are outlined below.

Page 6, lines 125-134: the whole genome of *Streptomyces atratus* was sequenced but the genome data is not reported except for the *ila* BGC. Do the authors intend to publish the genome elsewhere? It would be interesting to know how many BGCs were identified and the type of BGCs, along with information on assembly, contig number, etc.

Page 10, lines 210: "production of ilamycins was fully restored in the fed Δ ilaD mutant (Fig. 3, trace x)". The production was not fully restored, i.e. not to the same extent as in the WT. In fact,

the ratio of analogs obtained is different than for the WT, i.e. compound 5 is the main component and there appears to be only trace amounts of the other compounds. Could the authors comment on that?

Page 10, lines 210: "production of ilamycins was fully restored in the fed Δ ilaD mutant (Fig. 3, trace x) but not in the Δ ilaE mutant fed with 2,4-HDA or 4-HA (Fig. 3, traces xi and xii)". It should say "in the Δ ilaD mutant fed with L-AHA". Also, I'd suggest feeding L-AHA to the Δ ilaE mutant as positive control and as an attempt to chemically complement the mutant, since replacement of ilaE with the apramycin marker may have affected the expression of downstream genes.

Page 10, lines 212-214: "The above results indicate that IlaD performs α -oxidation to form α -keto intermediate 10, and that the sole aminotransferase within the cluster, IlaH, is responsible for transamination of 10 to form the L-AHA unit" A piece of data missing here is the result of feeding 4-HA to the Δ ilaD mutant.

Page 10, lines 214-216: "The oxidation of the C6 polyketide chain by IlaD and subsequent transamination at the α -position occurs while the substrate is tethered to the ACP3 of IlaE (Fig. 2e)". What evidence do we have of that?

Page 11, lines 237: "Silica analysis". In silico analysis is probably meant here.

Page 12, lines 280-282: "Since the Δ ilaL mutant was found to only generate compounds 1 and 2, the latter of which has an epoxy group, it can be readily determined that IlaL is responsible for the six electron L-Leu oxidation" I'd say the results indicate that ilaL is not essential for epoxide biosynthesis, and that by elimination (not direct evidence!) ilaL is a candidate for L-Leu oxidation.

Page 13, line 308: I suggest deleting "definitely".

Page 15, line 336: I suggest spelling out G+ and G- as "Gram-positive" and "Gram-negative".

Table 1: Were the anti-mycobacterial assays done in replicates? If so, please indicate the number of replicates and the standard deviation, if applicable. The statement that compounds 6 and 7 are exciting anti-TB drug leads should not be based on N=1.

I'd like to commend the authors for the thoroughness of their SI section.

Responsive letter to reviewers

Reviewer #1 Comments:

Reviewer #1 (Remarks to the Author):

In this interesting study, a family of compounds from marine bacterium *Streptomyces atratus* was analyzed, revealing this strain to be a producer of the known compounds known as ilamycins (rufomycins). Follow up structural analysis completed the determination of absolute stereochemistry for this family, identified and validated the gene cluster and facilitated the identification of some analogs in the series. In all 12 mutants and 5 chemical complementation experiments were performed for functional analysis and 10 additional replacements defined cluster boundaries.

Some proposals of the biosynthesis were generated based on gene replacement and chemical complementation studies. The nitration system is highly analogous to thaxtomin biosynthesis, and is as one would expect for this compound (at an activated position ortho to phenol). The polyketide synthase has some unusual features, especially a missing AT domain (see comment below). These studies, which complete the structure determination of a known series and inaugurate work into the biosynthesis are well executed, carefully described, and generally interpreted soundly (See comments below).

The genuinely surprising result from these studies is the observation that the analogs that accumulate because of disruption of an epoxidase (E1/E2) have specific nanomolar activity against tuberculosis, which are over 100-fold more active than closely related analogs, and substantially more potent than benchmark compounds. This is surprising because the electrophilic epoxides (C1/C2) are much less active than the terminal alkene E1/E2, and the aminal is only active in the presence of the terminal alkene. It appears from these data that both the aminal and the terminal alkene are essential. Given the clinical significance of tuberculosis, the potency of these newly elucidated compounds, and modest cytotoxicity a range of human cell lines (400 - 1900 fold less active in mammalian lines), these compounds could be important lead compounds in the development of new therapeutics. This result should be of interest to readers of Nature communications.

Comments:

1A. It is hard to parse this sentence: “All the binding pocket sequences of the A domains in IlaS do not match their predicted amino acid substrates, indicative of the

diversity of A domains and supporting the potential of their application in generating “engineered” natural products.” Firstly, it is typical for automated predictions of A-domain selectivity to yield ambiguous results. Secondly, that a homology model or neural network algorithm is unable to infer a prediction has no connection to biosynthetic plasticity or engineering potential.

Response: We appreciate this excellent point by reviewer #1. In accordance with their suggestion, we have revised the initial sentence in Page 7: to read **“PKS/NRPS analyses employing online software revealed that the predicted substrate amino acids of the A1-A7 binding pocket domain of IlaS do not match those found in the ilamycins. This disparity between predicted and actual (ilamycin) IlaS substrate structures suggests a high degree of substrate promiscuity in IlaS A domains and supports the potential of A domains in generating “engineered” unnatural NRPS natural products”**. This logic is supported by the well-known use of A domain specificities and replacement strategies across biosynthetic gene cluster to generate specifically modified natural products; daptomycin engineering efforts very elegantly highlight this kind of strategy (Nguyen KT., *et al.* Combinatorial biosynthesis of novel antibiotics related to daptomycin, *Proc. Natl. Acad. Sci. USA.* 2006, 103, 17462-17467).

1B. RE: the missing AT domain, have the authors gone beyond Antismash for domain analysis? Antismash, which is unsupervised and overly reliant on Markov models, is not sufficient for this purpose. The presence/absence of a divergent AT domain should be ascertained by BLAST analysis as well. The presence/absence of an AT-like sequence between KS₂/DH₁ should be confirmed and commented on.

Response: We appreciate this excellent point by reviewer #1. In accordance with their suggestion, we have analyzed the amino acid sequences between KS₂/DH₁ with NCBI protein BLAST online software and have found that the sequence has 34% identity (Similarity=43%) with an acyl transferase domain-containing protein (Sequence ID: SCL26451.1) in *Micromonospora pallida*. This may indicate that a divergent AT domain (an AT-like domain) is present in the KS₂/DH₁. The sequence alignment of the AT-like (AT₁-L) domain and a canonical AT (AT₂) domain in IlaE with other AT sequences revealed that the conserved S and R active residues in the GX₂SXGE...R motif were mutated (Supplementary Fig. 31) and the malonyl-CoA specificity motif (HAFH) were absent in this AT₁-L domain (Petković, H. *et al.* Substrate specificity of the acyl transferase domains of EpoC from the epothilone polyketide synthase. *Org. Biomol. Chem.* 6, 500-506 (2008)). This indicates the AT₁-L domain is not functional. The alignment results can be seen in Supplementary Figure 31 in Page S18. And we

have added an AT₁-L domain in Figure 2e. Accordingly, we have revised the sequence in Page 8 to “***i) an AT-like (AT₁-L) domain and a canonical AT (AT₂) domain assignable to IlaE, but the conserved S and R residues in the GX SXGE...R motif and the malonyl-CoA specificity motif HAFH are present only in the later canonical AT (AT₂) domain, suggesting that the later canonical AT₂ domain maybe used iteratively to synthesize the C6 chain.***”

1C. Error in Figure 2e IlaE, should be a KSQ (superscript Q) in the domain. Probably a graphics software glitch.

Response: We appreciate this excellent point by reviewer #1. We have revised the KSQ to KS^Q in Figure 2e IlaE.

1D. Paragraph starting 212 needs some qualification. The conclusions are consistent with the chemical complementation data, but in the absence of biochemical experiments, still preliminary. For instance, transport issue may affect uptake of precursors.

Response: We appreciate this excellent point by reviewer #1. We have revised the sentence in question to read: “**Based on the above feeding results and combined with the gene knockout results using *ilaDEH*, we propose that *IlaD* performs an α -oxidation to form α -keto intermediate 10, and that the sole aminotransferase within the cluster, *IlaH*, is responsible for transamination of 10 to form the *L-AHA* unit.**” (Page 10)

1E. Paragraph 309. This is paragraph facile, it is completely expected that the animals would interconvert as they are in equilibrium the aldehyde. Same phenomenon observed in anthramycin, for instance. **6/7** are epimers from simple animal equilibration hence are really one compound. This is typical and expected, for instance, see anthramycin.

Response: We appreciate this point by reviewer #1. The interconversion of **6** and **7** described in Page 14 was based on our HPLC analysis results. In order to show the accurate interconversion feature of **6/7**, we have added the HPLC results showing the interconversion of purified **6** and **7** in Supplementary Figure 34A in Page S21. In addition, the Supplementary figure 34A was also noted in the main text.

1F. It would improve the manuscript to have a little more understanding for the remarkable activity of **6/7**. It is possible that the epoxide is unstable under assay conditions. It may open to the vicinal-diol via hydrolysis, resulting in a structural

change that decreases activity. Have the authors checked the stability of the epoxide under bioassay conditions by HPLC/MS?

Response: We appreciate this concern pointed out by reviewer #1. We have checked the stability of epoxides **2**, **3/4** and **5**; HPLC analyses and LC-MS revealed that, under bioassay conditions, these compounds undergo little to no change with respect to molecular weights, UV spectrum and HPLC retention times. On the basis of these findings we theorize that the terminal alkene and carboxylation at C₃₃ are main criteria dictating the unique activity of compounds **6/7**.

Reviewer #2 Comments:

Reviewer #2 (Remarks to the Author):

Ma et al describes the identification of the ilamycin biosynthetic gene cluster from a *Streptomyces atratus* strain with rare L-3-nitro-tyrosine and L-2-amino-4-hexenoic acid (L-AHA) structural features. By a series of gene inactivation and feeding experiments with precursors including isotope-labeled precursors and subsequent structural elucidation of the accumulated intermediates, they clarified the biosynthesis steps and involved genes for L-3-nitrotyrosine and L-AHA, as well as the order of tryptophan prenylation and isoprene epoxidation.

This work sounds solid and the manuscript is well written and can be easily followed.

I recommend it for acceptance with minor revision.

Minor points Lines 216 and 217

2A. The authors concluded the release of L-AHA from the PKS before being loaded onto the NRPS assembly line, based on the feeding experiment with AHA. Can they exclude the possibility that the observed results with AHA represent just an artificial effect?

Response: We thank reviewer #2 for raising this question. Firstly, when we fed L(D)-AHA to two $\Delta ilaD$ mutants (twice) and the $\Delta ilaE$ mutant, all of them were found to produce ilamycins and the authenticity of the compounds was confirmed by HRMS. Secondly, when we fed 2,4-HDA and 4-HA to the $\Delta ilaE$ and $\Delta ilaD$ mutants, the production of ilamycins wasn't restored in both mutants. As a result of these findings we believe we can exclude the possibility that the results observed with AHA represent a simple artificial effect.

2B. Lines 237 and 258: Silica analysis?

Response: We truly appreciate this excellent catch by reviewer #2 and have revised

the “Silica analysis” to “Silico analysis” (page 11).

Reviewer #3 Comments:

Reviewer #3 (Remarks to the Author):

The unusual structure of ilamycins, containing rare L-2-amino-4-hexenoic acid (L-AHA) and L-3-nitro-tyrosine structural motifs makes them interesting starting points for biosynthetic studies. In addition, two ilamycin congeners show potent anti-tuberculosis activity (MIC values in the low nanomolar range), which makes them biomedically relevant as well.

The manuscript by Ma et al. is a comprehensive study of ilamycins, describing 1) the genome sequence of the producing bacterium (*Streptomyces atratus*); 2) the identification of the ilamycin biosynthetic gene cluster (BGC); followed by 3) the generation of 22 deletion mutants that allowed the authors to delineate the borders of the BGC and to probe the function of structural genes; 4) the absolute stereochemistry of ilamycin congeners; 5) insights into the biosynthesis of the unusual building blocks L-AHA and L-3-nitro-tyrosine; and 6) biological activity of ilamycins.

My recommendation is to publish the manuscript after minor revisions. Specific comments/recommendations are outlined below.

3A. Page 6, lines 125-134: the whole genome of *Streptomyces atratus* was sequenced but the genome data is not reported except for the *ila* BGC. Do the authors intend to publish the genome elsewhere? It would be interesting to know how many BGCs were identified and the type of BGCs, along with information on assembly, contig number, etc.

Response: We appreciate this concern pointed out by reviewer #3. The whole genome sequencing of *S. atratus* SCSIO ZH16 was finished (0 gap) with a combination of 2nd-generation 454 and 3rd-generation PacBio sequencing technology. The size of the entire linear genome (1 contig) of *S. atratus* SCSIO ZH16 is 9.64 Mbp; antismash annotation results demonstrated that there are 26 biosynthetic gene clusters encoded on its genome. The 30 gene clusters can be associated with seven families of compounds: 3 NRPS, 3 PKS, 4 PKS/NRPS, 3 Terpene, 2 siderophore, 3 bacteriocin clusters and 8 other BGCs. Just as reviewer #3 has suggested, these data, combined with other results forthcoming, will be submitted shortly to *J. Biotech.*

3B. Page 10, lines 210: “production of ilamycins was fully restored in the fed Δ *ilaD*

mutant (Fig. 3, trace x)". The production was not fully restored, i.e. not to the same extent as in the WT. In fact, the ratio of analogs obtained is different than for the WT, i.e. compound **5** is the main component and there appears to be only trace amounts of the other compounds. Could the authors comment on that?

Response: We appreciate this insight and concern as raised by reviewer #3. We had not given enough thought to the use of the word "fully". In light of reviewer 3's comment, we have revisited this idea and have correspondingly removed the word "fully" from this sentence (page 10, lines 210). See below section 3C for a more precise description of how this discussion has been rephrased to be more accurate, in part, by the removal of "fully" as well as by our efforts to reflect reviewer 3 suggestions section 3C).

In regards to concerns about compound **5**, we note that **5** is the most fully oxidized of the eight ilamycin congeners identified herein. Also worth keeping in mind is that the rate and timing of precursor biosynthesis as well as the timeliness of post-tailoring enzyme expression/activation likely determines the ratios and absolute titers of the ilamycin congeners noted. Consequently, in the feeding experiments carried out, compound **5** is very likely the main observable component due to possible precursor positive regulation. We envision that this would trigger enhanced transport and/or incorporation of such precursors to the ilamycins assembly line and thus, enhanced rates of final product generation.

3C. Page 10, lines 210: "production of ilamycins was fully restored in the fed $\Delta ilaD$ mutant (Fig. 3, trace x) but not in the $\Delta ilaE$ mutant fed with 2, 4-HDA or 4-HA (Fig. 3, traces xi and xii)". It should say "in the $\Delta ilaD$ mutant fed with L-AHA". Also, I'd suggest feeding L-AHA to the $\Delta ilaE$ mutant as positive control and as an attempt to chemically complement the mutant, since replacement of *ilaE* with the apramycin marker may have affected the expression of downstream genes.

Response: We appreciate this weakness pointed out by reviewer #3. In accordance with the suggestions of reviewer #3, the feeding experiment with L-AHA to $\Delta ilaE$ mutant was carried out. Notably, feeding of L-AHA to the $\Delta ilaE$ mutant strain restored ilamycin production. Based on these results and four other feeding experiments using different precursors, we have proposed that L-AHA is first released from the PKS assembly line and then incorporated into the NRPS assembly line to fulfill the biosynthesis of ilamycins. The feeding results of $\Delta ilaE$ with L-AHA were added to Fig 3 in trace xii. And we have revised the "HPLC analysis of the fermentation extract revealed that the production..... in page 10" to "***HPLC analyses of the fermentation extracts revealed that ilamycin production was restored in both the $\Delta ilaD$ and $\Delta ilaE$ mutants when supplied with L(D)-AHA (Fig. 3, traces x, xii).***

However, neither mutant strain produced ilamycins when supplied with 4-HA and feeding 2,4-HDA to the $\Delta ilaE$ mutant also failed to restore ilamycin production (Fig. 3, traces xi-xiv)."

3D. Page 10, lines 212-214: "The above results indicate that IlaD performs α -oxidation to form α -keto intermediate **10**, and that the sole aminotransferase within the cluster, IlaH, is responsible for transamination of **10** to form the L-AHA unit" A piece of data missing here is the result of feeding 4-HA to the $\Delta ilaD$ mutant.

Response: We appreciate this excellent point by reviewer #3. In response to the suggestion by reviewer #3, 4-HA was also fed to the $\Delta ilaD$ mutant (Page 10), and these feeding results and other feeding results indicated that IlaD performs α -oxidation to form α -keto intermediate **10**, and that the sole aminotransferase within the cluster, IlaH, is responsible for transamination of **10** to form the L-AHA unit. The feeding results of $\Delta ilaD$ with 4-HA were added in Fig 3 as trace xi.

3E. Page 10, lines 214-216: "The oxidation of the C6 polyketide chain by IlaD and subsequent transamination at the α -position occurs while the substrate is tethered to the ACP3 of IlaE (Fig. 2e)". What evidence do we have of that?

Response: We appreciate this concern by reviewer #3. As mentioned in the main text, inactivation of *ilaD* and *ilaE* both abolished the production of ilamycins. Feeding L-AHA to the $\Delta ilaD$ mutant restored the production of ilamycins, while feeding 4-HA to the $\Delta ilaD$ mutant could not restore the ilamycins production. These data suggest the oxidation of the C6 polyketide chain by IlaD and subsequent transamination at the α -position occurs while the substrate is tethered to the ACP₃ of IlaE.

3F. Page 11, lines 237: "Silica analysis". In silico analysis is probably meant here.

Response: We really appreciate this excellent catch by reviewer #3 and have revised the "Silica analysis" to "Silico analysis" (page 11).

3G. Page 12, lines 280-282: "Since the $\Delta ilaL$ mutant was found to only generate compounds **1** and **2**, the latter of which has an epoxy group, it can be readily determined that IlaL is responsible for the six electron L-Leu oxidation" I'd say the results indicate that *ilaL* is not essential for epoxide biosynthesis, and that by elimination (not direct evidence!) *ilaL* is a candidate for L-Leu oxidation.

Response: We appreciate this excellent catch by reviewer #3. We have revised the sentence: "Since the $\Delta ilaL$ mutant was found to only generate compounds **1** and **2**, the latter of which has an epoxy group, it can be readily determined that IlaL is responsible for the six electron L-Leu oxidation" (page 13) to read as follows: "Since

the *AilaL* mutant was found to generate only compounds 1 and 2, the latter of which has an epoxy group, we propose that IlaL is not essential for epoxide biosynthesis. Rather, deductive reasoning and by process of elimination, it appears that IlaL plays a central role in the net 6-electron oxidation of L-Leu to install the lone carboxylic acid observed in certain ilamycins.”

3H. Page 13, line 308: I suggest deleting “definitely”.

Response: We appreciate this excellent point by reviewer #3 and have accordingly deleted “definitely” in line 308 (Page 14).

3I. Page 15, line 336: I suggest spelling out G⁺ and G⁻ as “Gram-positive” and “Gram-negative”.

Response: We appreciate this concern pointed out by reviewer #3 and have changed G⁺ and G⁻ to read “Gram-positive” and “Gram-negative” in Page 15 and Page 24.

3J. Table 1: Were the anti-mycobacterial assays done in replicates? If so, please indicate the number of replicates and the standard deviation, if applicable. The statement that compounds 6 and 7 are exciting anti-TB drug leads should not be based on N=1.

Response: The anti-mycobacterial assays were performed in triplicate using the methods described in the reference (Zhang, T; Li, S-Y.; Nuermberger, E. L. *Plos One*, **2012**, 7 e28744.) According to the standard methods provided by Clinical and Laboratory Standards institute (CLSI), the MIC means “the lowest concentration of antimicrobial agent that completely inhibits growth of the organism in the tubes/plates or microdilution wells as detected by the unaided eyes”. Therefore, we think the deviations have no meaning in the test. We have added the notation that "N=3" in Table 1 and add “All experiments were performed in triplicate with kanamycin and rifampicin as control agents” in page 24.

3K. I’d like to commend the authors for the thoroughness of their SI section.

Response: We really appreciate this constructive feedback from reviewer #3.

REVIEWERS' COMMENTS:

Reviewer #1 (Remarks to the Author):

The authors have responded comprehensively to the concerns posed in review with the following exception. This sentence:

"This disparity between predicted and actual (ilamycin) IlaS substrate structures suggests a high degree of substrate promiscuity in IlaS A domains and supports the potential of A domains in generating "engineered" unnatural NRPS natural products"

This still represents a misinterpretation of the homology modeling of A-domains. As written the sentence asserts that as the specificity code does not match anything in the databases, it must be promiscuous. There is no connection. The fact that A-domains selectivity has been modified in the past (and relatively few have to result in new natural products) is non-sequitur to this argument. I recommend removing this sentence, it detracts from the rest of the work.

Reviewer #2 (Remarks to the Author):

My questions have been answered and the manuscript can be now accepted for publication

Reviewer #3 (Remarks to the Author):

The manuscript has been satisfactorily revised. Only two minor comments remain that I believe may be addressed during editorial review, if the authors agree.

3E. [revised manuscript page 10, line 224]: I suggest adding "likely" in the following sentence: subsequent transamination at the α -position likely occurs while the substrate is tethered to the ACP3 of IlaE"

1A. I agree with following comment by reviewer 1 "that a homology model or neural network algorithm is unable to infer a prediction has no connection to biosynthetic plasticity or engineering potential" and would, therefore, suggest deleting the sentence "This disparity between predicted and actual (ilamycin) IlaS substrate structures suggests a high degree of substrate promiscuity in IlaS A domains and supports the potential of A domains in generating engineered unnatural NRPS natural products" [page 7, lines 141-144].

Responsive letter to reviewers

Reviewer #1 Comments:

Reviewer #1 (Remarks to the Author):

The authors have responded comprehensively to the concerns posed in review with the following exception.

1A: This sentence: "This disparity between predicted and actual (ilamycin) IlaS substrate structures suggests a high degree of substrate promiscuity in IlaS A domains and supports the potential of A domains in generating “engineered” unnatural NRPS natural products"

This still represents a misinterpretation of the homology modeling of A-domains. As written the sentence asserts that as the specificity code does not match anything in the databases, it must be promiscuous. There is no connection. The fact that A-domains selectivity has been modified in the past (and relatively few have to result in new natural products) is non-sequitur to this argument. I recommend removing this sentence, it detracts from the rest of the work.

Response: We appreciate this concern pointed out by reviewer #1 and have removed this sentence in the newly revised manuscript, page 7 paragraph 2.

Reviewer #2 Comments:

Reviewer #2 (Remarks to the Author):

My questions have been answered and the manuscript can be now accepted for publication.

Response: We really appreciate this constructive feedback from reviewer #2.

Reviewer #3 Comments:

Reviewer #3 (Remarks to the Author):

The manuscript has been satisfactorily revised. Only two minor comments remain that I believe may be addressed during editorial review, if the authors agree.

3A. [revised manuscript page 10, line 224]: I suggest adding “likely” in the following sentence: subsequent transamination at the α -position **likely** occurs while the substrate is tethered to the ACP₃ of IlaE”

Response: We appreciate this concern pointed out by reviewer #3 and have added “likely” in the newly revised manuscript, page 10 paragraph 2, line 221 to read as “subsequent transamination at the α -position likely occurs while the substrate is tethered to the ACP₃ of IlaE.”

3B. I agree with following comment by reviewer 1 “that a homology model or neural network algorithm is unable to infer a prediction has no connection to biosynthetic plasticity or engineering potential” and would, therefore, suggest deleting the sentence “This disparity between predicted and actual (ilamycin) IlaS substrate structures suggests a high degree of substrate promiscuity in IlaS A domains and supports the potential of A domains in generating engineered unnatural NRPS natural products” [page 7, lines 141-144].

Response: We appreciate this concern pointed out by reviewer #3 and have deleted this sentence in the newly revised manuscript, Page 7 paragraph 2.

We greatly appreciate the constructive input of all reviewers and hope they recognize that their contributions have greatly strengthened the manuscript.